# Environmental and Socio-Economic Effects of Underground Brown Coal Mining in Piła Młyn (Poland)

**Mirosław Rurek [1,*][ID], Alicja Gonia [2] and Marcin Hojan [1][ID]**

[1] Department of Landscape Geography, Institute of Geography, Kazimierz Wielki University, Kościeleckich Square 8, 85-033 Bydgoszcz, Poland; homar@ukw.edu.pl

[2] Department of Socio-Economic Geography and Tourism, Institute of Geography, Kazimierz Wielki University, Kościeleckich Square 8, 85-033 Bydgoszcz, Poland; alagon@ukw.edu.pl

* Correspondence: mirur@ukw.edu.pl

**Abstract:** In Poland, apart from opencast mining, brown coal (lignite) was also mined by underground methods. This is related to glaciotectonic disturbances leading to deposition of Miocene coal in the form of folds (synclines and anticlines). The highest number of underground brown coal mines in the 19th century was recorded in western Poland. In northern Poland in the second half of the 19th century there were active underground brown coal mines in Piła Młyn. The study aims to present the environmental and socio-economic effects of discontinuing lignite mining. It is a unique example of cultural heritage and influences the tourist development of the region. To this end, historical topographic maps were used together with data from LIDAR (Light Detection and Ranging) laser scanning, available descriptions and scientific articles about coal mines. Information from the local inhabitants and representatives of the "BUKO" Association (the Association of Inhabitants and Enthusiasts of Piła nad Brdą "BUKO") was also obtained. As shown by the results of the analyses, changes in the environment are manifested in terrain relief and highlight the location of former coal mines. Clear land subsidence is arranged in linear sequences and occurs as single landforms. Recesses at the former extraction sites are very distinct. The socio-economic effects observed include activity of the inhabitants and development of rural areas. The area also offers good prospects for tourism as a themed mining village attracts tourists and is a unique amenity in Poland. The subject matter of this article is also of importance from the standpoint of civil engineering since post-mining areas cannot be repurposed for residential development. Due to changes in the relief in this area in places other than those analyzed, scientific research will be conducted here, which will allow to answer other hypotheses (brown coal transport).

**Keywords:** land subsidence; recesses; underground lignite mine; themed village; Tuchola forest

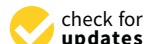



## 1. Introduction

Brown coal is a natural, non-renewable resource used for generating electricity and heat. It originated during decay of organic matter deposited in depressions. A material factor in the peat-formation process was the water table that gave rise to anaerobic conditions. Younger plants dying formed successive layers impacting the underlying strata, at the same time increasing pressure and temperature [1,2]. Deposits of brown coal date back to different geological periods. In Poland brown coal originated in the period from the Triassic to the Paleogene and Neogene [2].

Brown coal mining in Poland is preconditioned by glaciotectonic disturbances that occurred when the Scandinavian ice sheet reached Poland. Brown coal, deposited at considerable depths, was extruded towards the surface during the Middle Polish glaciation [3]. Such examples can be found in western Poland [4]. In other locations (central Poland—Konin, Bełchatów) coal deposits are overlaid with glacial sediments from 30 m to 300 m thick [2].

In Poland brown coal is mainly extracted in opencast mines. The history of brown coal extraction in Poland dates back to the 17th century when coal was mined underground at the border of Poland, Czechia and Germany [2]. Opencast mining has a long tradition in Poland. As early as 1905 the first large opencast mine, Herkules, was established. It now exists as the Turów Coal Mine. Deposits of brown coal in western Poland are associated with glaciotectonic disturbances [5]. In 1873 the Sieniawa mine was established which operates to this day. It should be emphasized that at the initial stage coal was extracted by underground mining methods there. This is significant, since its present extraction leads to the formation of huge pits altering the environment. The effects of such activity can be observed within the area of the coal reserves named after nearby localities. These are Adamów, Koźmin, Bełchatów—Pole, Bełchatów, Bełchatów—Pole Szczerców, Pątnów IV, Drzewce, Tomisławice, Turów and Sieniawa 2 [2].

Extraction of hard and brown coal contributes to changes in the natural environment. Its exploitation is performed in open casts and deep mines. Deep mining tends to result in land subsidence and produces considerable waste heaps. Open casts, on the other hand, necessitate establishing deep excavations over a considerable area. Moreover, open-cast excavations lead to a loss of arable land, air and water pollution, deforestation, decrease in biodiversity and may cause mass movements on waste heaps and cast slopes [6]. In this respect, studies investigating the impact of open-cast excavation on soil are particularly note-worthy. Feng et.al. [7] reviewed available literature to demonstrate changes in physical, chemical and biological properties of soils in the vicinity of open-cast brown coal excavation sites. The analysis of scientific works carried out by [7] shows that changes in the soil in the area of open-cast mines are still not sufficient. Research on chemical properties, recultivation of post-mining sites and new research related to ecological aspects must be carried out. The impact of open-cast brown coal excavation on the environment and subsequent remediation measures were also extensively discussed in the context of Germany [8]. More recent studies conducted in that country employed digital terrain models to account for long-term influence over the period of 100 years [9,10]. Furthermore, open-cast brown coal mining operations in Rhineland (Germany) were found to exert considerable influence on socio-economic relationships, which is mostly attributed to resettlements and large-scale changes in road infrastructure [11].

Underground coal mining entails vast environmental changes in the affected urban areas. Hard coal mining in the Silesian Uplands results in land subsidence. Geomorphological and hydrological consequences manifest in the occurrence of wet meadows and subsidence basins that are filled with water and constitute surface water reservoirs [12–16]. Said reservoirs accumulate sediments which can be analyzed for the presence of various chemical elements. [13,15]. Other examples of environmental changes were presented in studies pertaining to the town of Wałbrzych (SW Poland), which features a former hard coal excavation site. The aforementioned works present analyses of geomorphological forms related to the cessation of coal exploitation, including subsidence basins, waste heaps and settling ponds [17]. Jancewicz et.al. [18] extended their analyses to account for linear element such as trenches and railroad embankments.

The transformation of relief is also related to the superstructure of surface in urban and rural areas, not only post-mining. Spatial data analyses constitute a particularly note-worthy method for demonstrating changes within urban areas. Historical data and digital models, on the other hand, can be used to indicate the impact of excavation on land relief in the course of anthropogenic transformations both in terms of surface [19] and linear changes [20,21].

Underground coal mining is rare but some information regarding such a method of extraction can be found [1,4,5,22]. Layers of brown coal have a different thickness, are very often folded and arranged as vertical outcrops. Brownfields display traces of underground human activity in the form of land subsidence. Subsidence basins as negative landscape forms may also be related with natural factors. Contemporary and Holocene impacts of natural factors on land relief have been discussed in a number of academic publications.

The authors indicate collapsed beaver burrows as one of such naturally-occurring land relief elements [23–26].

This paper aims to present the impact of former underground brown coal mining in Piła Młyn on the natural environment and the possibility of using the effects of operating underground adits visible on the terrain surface for tourism. In pursuit of the above-mentioned objective the authors attempted to address the following questions: How did the cessation of mining operations transform the natural environment? How did it impact the local population and its activation? Thus far the extent of environmental changes related to the impact of former mining operations in this area has not been analysed in such detail. In fact, this example may be considered unprecedent in the scale of Poland and perhaps even in the scale of Europe.

## 2. Methods and Study Area

### 2.1. Location of Study Area and Geology

The study site is situated in the village of Piła Młyn in Kuyavian-Pomeranian voivode-ship in Poland (Figure 1). In terms of administrative division the village of Piła Młyn is located in the commune of Gostycyn and has a population of 130 inhabitants. As far as geophysical division is concerned, Piła Młyn lies in the mesoregion of the Brda Valley, which is a part of the Central European Plain [27]. It is located between the Brda River valley (on the right bank) in the southeast and Lake Szpitalne in the northwest (Figure 2). At this point the Brda River undercuts the shore and forms 20 m tall steep slopes. The area can be split into a northern and southern part, as it is intersected by the road to Gostycyn. It extends over about 100 ha. Its surface displays subsidence pits of various sizes that were clearly formed as a result of underground mining. The area is covered with a pine forest growing on poor-quality soil that originated from mineral sediments. These are fluvioglacial sediments building the Brda river outwash plain formed by glacial meltwater flowing southwards during the Pomeranian phase of the Vistulian (Weichselian) glaciation. The outwash plain's sediments have a different thickness ranging from 10 to 20 m.

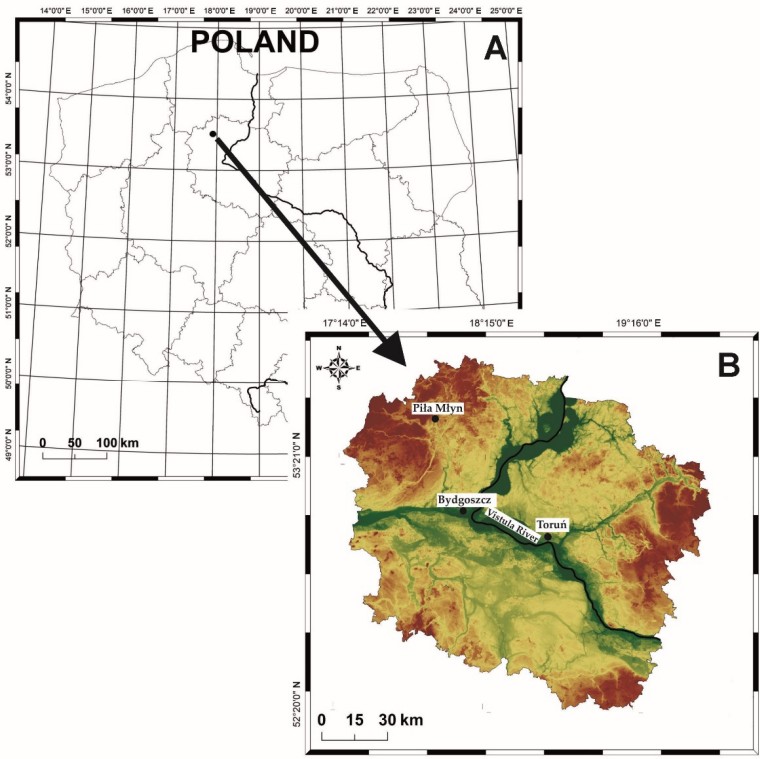

**Figure 1.** Location of the research area against the background of Poland (**A**) and the Kuyavian-Pomeranian Voivodeship (**B**).

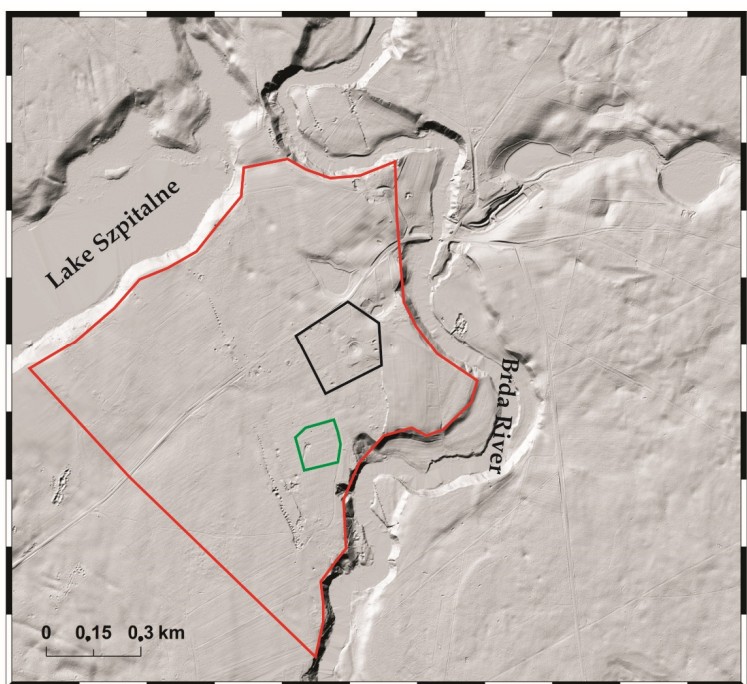

**Figure 2.** Post-mining areas in Piła Młyn. 1—area with brown lignite deposits (red marking), 2—residential area (black color), 3—area of tourist attraction "Mining Village" (green color), source: [28].

*2.2. Research Methods*

This paper makes use of available information on the stages of operation of coal mines in Piła Młyn. The environmental consequences were presented using Light Detection and Ranging (LIDAR) data [28], archival maps and the results of field studies. Analysis of these sources allowed identifying potential hazard sites and potential tourist areas (western shore of the Brda River Valley). Laser scanning data was compared against historical maps available online [29]. Prussian maps were used for analysing terrain changes. Other inputs were data from Polish Central Geological Database (CBDG) [30] and geological maps in the scale 1:50,000. All comparative analyses were processed with the ArcGis 10.1 software (ESRI, Redlands, CA, USA) using external Web Map Service (WMS) connections. The location of the mines at the time of their operation was captured based on available historical data including location sketches and layouts of extraction fields. They were georeferenced with ArcGis 10.1 software in order to identify changes in the environment.

The analysis of socioeconomic consequences was predicated on various sources. The socio-economic effects were presented using information provided by the "BUKO" Association, as well as information from press and websites. Moreover, valuable data were obtained through interviews with the representatives of the Buko association and the inhabitants of the village. The main tourist attractions were presented along with various measures undertaken by members of the "BUKO" Association in order to stimulate the activity of the local population and revitalise tourism in the area of the historic coal mine.

**3. Results**

*3.1. History of Brown Coal Mines in Poland*

The oldest underground brown coal mine in Poland is "Caroline" in Glisno near Lubniewice. It received a mining licence in 1820. In the 1940s, in Lower Silesia and Lusatia, brown coal was extracted mainly by small mines [31]. The extraction volume in respective mines was primarily associated with the accessibility of brown coal reserves.

Many inactive underground brown coal mines exist in the Lubusz Region [32–36]. Respective mines extracted brown coal at different times—from the mid-19th century

until the late 1950s. Coal mines declined due to damage caused by World War II, lack of professional human resources, technical issues and the exhaustion of coal reserves [31].

In the present Kuyavian-Pomeranian voivodeship, near Piła Młyn, there are remnants of brown coal mines. The reserves of brown coal were exposed due to lateral erosion of the Brda River. Around 1850 the inhabitants of Piła Młyn took advantage of this and started mining brown coal to heat their houses [1]. As late as 1892 the first brown coal mine in that area—Buko—was put into operation. In total six brown coal mines were established near Piła Młyn: Buko, Olga, Montania, Aleksandra (Alexandra), Zofia, and Teresa (Barbara) [1].

The mines needed accompanying infrastructure such as a narrow-gauge railway line connecting the Buko mine to Tuchola and the Olga mine to Gostycyn. In this area coal was extracted underground. Coal deposits, depending on the extraction site, were from 1.0 to 4.0 m thick and their slope ranged from 20 to 35°. Brown coal deposits were extracted from a depth of up to 30 m b.g.l. (below ground level). Respective mines operated at different times and were opened and closed; for instance, the Olga mine that operated in 1898–1911, 1917–1922 and 1932–1934. This was due to legal issues and technical obstacles. Coal mines in the area of Piła Młyn were ultimately closed during World War II [1].

In 2007 the Association of Inhabitants and Enthusiasts of Piła nad Brdą "BUKO" was established to conduct research on the history of the village. This led to the acquisition of archival photographs and identification of the remains of the mines by the inhabitants. Further work in archives provided more information. In addition, archaeological surveys involving a group of volunteers led to recreating the remains of the Montania coal mine, including the Olga drift mine [1].

Brown coal mining caused land subsidence even during operation of the mines, so some shafts were closed. Subsidence pits in the area of the former coal mines in Piła Młyn show the direction of brown coal extraction (Figure 2).

*3.2. Environmental Consequences of Discontinuing Brown Coal Mining*

Extraction of brown coal leads to transformation of the natural environment. The greatest transformations of the environment are caused by opencast mining of coal which alters the whole area of the coal mine. These include pits and spoil heaps, degradation of vegetation, lowering of the level of disappearance of underground water, and changes in the levels of still water reservoirs and rivers. In the case of underground mining of brown coal, extraction leads to the formation of spoil heaps, changes in the groundwater table level and changes on the surface. During underground extraction of coal these changes are insignificant in comparison to opencast mining. Only after the mining stops, when the underground infrastructure is left unsecured, do subsidence pits form on the surface.

Operation of underground brown coal mines transformed the environment in a way that for several years has been making the lives of the local inhabitants difficult. Analysing the available cartographic materials from the 19th and 20th century (German topographic maps Messtischblatt), a designation of the Buko mine can be found on maps from 1905 (Figure 3A1) and the Olga mine—on maps dating back to 1930, 1933 and 1940 (Figure 3B1,C1,D1). The latter mark depressions formed by collapses of galleries closest to the surface of the Olga mine and the subsidence pit of the Buko mine. German maps provide a lot of details but some of them were prepared on base maps from 1887, which can be misleading when interpreting the changes (Figure 3A).

The available laser scanning LIDAR offers new possibilities for interpreting the transformations of the environment related to the coal mines in Piła Młyn. Its accurate data reflects all changes in terrain relief. Analysis of LIDAR data allowed associating terrain relief transformations to respective extraction sites [1]. The largest pits are situated in the northern part of the analysed area (Figure 4A) and have a linear NW-SE course. They cover a distance of about 450 m. They are round in shape, with a diameter of 40 m and depths reaching 2.5 m. The biggest subsidence pit is situated at the centre of the built-up area in the village of Piła Młyn. Only a few subsidence pits can be seen on the land surface. Their shape is attenuated, which implies that they have been a part of this terrain for a long time.

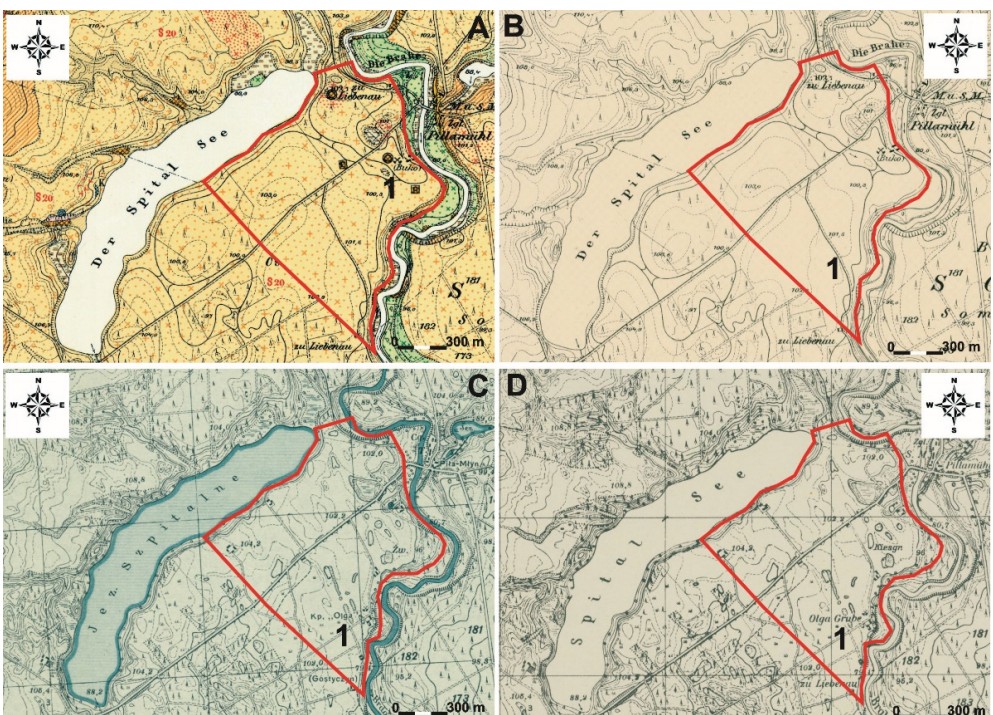

**Figure 3.** Historical topographic maps. (**A**)—German geologic-agricultural map from 1905, Tuchola sheet, (**B**)—German topographic map from 1930, Tuchola sheet, (**C**)—Polish topographic map from 1933, Bysław sheet, (**D**)—German topographic map from 1940, Tuchola sheet. **A**1—localization of Buko mine, **B**1–**D**1—localization of Olga mine, source [29].

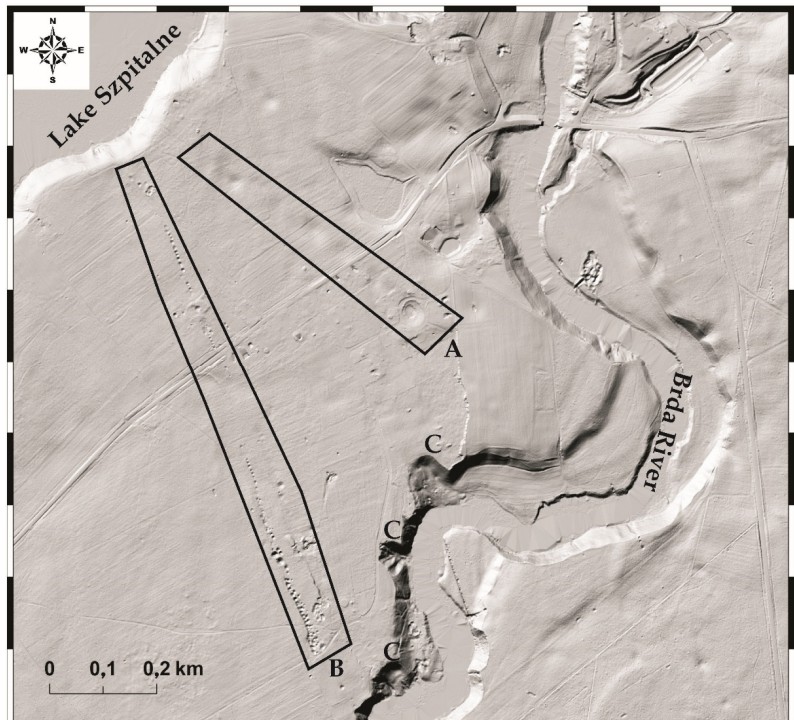

**Figure 4.** Detailed location of areas transformed by underground lignite mining. (**A**)—subsidence landforms after the Buko mine, (**B**)—subsidence landforms after the Olga mine, (**C**)—subsidence recesses. Source: [28].

Smaller but clearer subsidence landforms are situated between the Brda River Valley and Lake Szpitalne. They do not occur along, as they are intersected by the road to Gostycyn. Stretches on both sides of the road have a course similar to the layout of old subsidence pits but are more northward. The stretch starting from the Brda River Valley is made of 55 subsidence pits the biggest of which are reminiscent of circles. The two biggest have a diameter of 14 m and 12 m and are 3.5 m deep. The initial part of this stretch is an arrangement of more than ten pits constituting a strongly transformed terrain. The second stretch begins with pits by the road to Gostycyn. Its course displays well-developed pits that form a single big depression at the end of the subsidence area. The biggest pits are 8 × 8 m and 7 × 7 m in size and 2.5 m in depth. In addition to the above-discussed stretches of subsidence pits clearly related to mining activity, single pits occur that are a proof of hollows existing under the surface (Figure 4B).

The landforms that are clearly evident in the analysed area are subsidence recesses. Their origin is connected with mining activity as they are located at the outlets of underground galleries. They occur in the slopes of the Brda Valley at brown coal bed outcrops. Subsidence recesses are visible when comparing the German geological and agrarian map with the laser scanning image (Figure 4C). In the Brda River Valley, in the analysed area, there are only three recesses of different sizes. The first is 95 m wide and slope retreat is 65 m deep. The second recess is 70 m wide, and the slope cut reaches 45 m. The third recess is 75 m wide, and the slope retreat is 45 m. In all recesses, the height measured from the upper edge to the water table level in the Brda River is 18 m. At the outlets of the first and the third recess the terrain becomes more even and the sediments reach the channel of the Brda River. The second recess is in direct contact with the channel of the Brda River. An area at the upper edge before the third recess is worth mentioning. It is transformed with distortions visible on the laser scanning image. Supposedly, some sort of an excavation was buried there—possibly an access road to the recess.

### 3.3. The Use of Coal Mines for Tourism

3.3.1. The "BUKO" Association as an Initiator of Social Changes in the Area of Coal Mines in Piła-Młyn

At present, the area of the former brown coal mines in Piła-Młyn is mostly overgrown with a forest, and partially covered with individual buildings. The surface features numerous evidences of mining activity, concrete and brick casings of galleries, remains of buildings, posts, spoil heaps, steam pipelines and mining subsidence pits designating exhausted reserves [22]. More than one hundred archaeological sites were identified there [37]. For safety reasons, the underground part of the coal mines is inaccessible—both the shafts and adits were buried.

The area of the former coal mine is under the custody of the "BUKO" Association—its discoverer and initiator of the first archaeological survey and its development for tourism. The "BUKO" association comprises enthusiasts, inhabitants of the town of Piła nad Brdą. The association operates as a non-profit social enterprise [38]. The name "BUKO" comes from the name of the first brown coal mine established in the area by David and Jakob Bukowzer. The Mining Village established in that area by the "BUKO" Association, incorporating the remains of the only underground brown coal mine preserved in northern Poland, is a unique monument of technology [39], and the only such themed village and a unique amenity in all Poland [40].

The project "The Themed Mining Village as a Mine of Social Economics" allowed implementing projects within the premises of the former coal mine based on cultural resources of the area in order to conserve the monument and adapt it for tourism, stimulate the activity of inhabitants and improving the quality of their lives and disseminate knowledge about the cultural heritage of this part of the Tuchola Forest. Local resources and infrastructure in the settlement of Piła were catalogued. The needs of the local community and its interest in joint activities connected with establishing and running a themed village were analysed. In the first place the settlement of Piła was separated from Gostycyn (2008), and then the rank

of the village was raised by creating a village with its own budget (2011). In 2016, thanks to the efforts of the "BUKO" Association, the cultural landscape of underground brown coal mines and a part of the brownfield, including plots number: 147/12, 147/14, 880, 883/2, were entered into the register of monuments [41].

### 3.3.2. Amenities and Tourist Traffic in the Mining Village

At present, the village offers many amenities, including main ones—connected with the coal mine that is the theme of the village, and additional ones—making the offer richer. The reconstructed entry to the drift mine and relics of the engine house on the surface, including the steam boiler footing and a system of overflow tanks of the Montania mine, are available for sightseeing (Figure 5A,B). Organised role play tours of the mine are guided by the Coal Mine Spirit called Skarbnik (Treasurer). The mine can also be explored via a designated nature and history path with an audio-guide recounting the story of the coal mine and talking about the natural environment of the Tuchola Forest and outdoor games such as quest, Geocaching and strategy games matching the age of the participants (Figure 6A,B). An interactive model of the underground coal mine from 1900–1921 in the old engine house and a reconstructed mine shaft and headframe of the Montania mine dating back to 1917–1921 are also interesting (Figure 6C). The village offers a coal mining and old miner lamps lighting show. Additional amenities include a 19th century playground presenting old-time games and a sensory path covered with pine-cones, coal and moss. Wafer baking with coal is demonstrated. Numerous workshops are organised (including crafts, arts, cooking and nature, gardening, old-fashioned professions and workshops during which gala mining hats are created), along with study visits for non-governmental organisations, foundations, informal groups and public institutions, as well as conferences and seminars. Other popular events include team-building, the "Foreman Wedding" feast which reconstructs a traditional wedding typical of the region of the Tuchola Forest and Silesia in the 1930s, and St Barbara's Beer Fest. The "Old Sawmill" organises educational classes and meetings of the members of the Association and the village council [42]. Ultimately, this is going to be a Local Centre for Social Integration and Activity used for education and training of the inhabitants [43]. Another step towards stimulating the inhabitants to action was the establishment of a social enterprise called "Mining Village" that took over the service of tourist traffic and employed five professionally inactive inhabitants of the commune. The mining village plans to set up an interactive underground tourism trail presenting the 19th century lignite mining technology in an innovative manner, a museum of brown coal mining in the Tuchola Forest and a centre of education on the conservation of monuments, culture and local history. It aims to create a tourism product of the Kuyavian-Pomeranian voivodeship in the form of an open-air museum of underground brown coal mining unique in northern Poland [1]. The increasing number of tourists testifies to the popularity of the mining village. A big increase up to 3000 visitors was noted in 2013 [43]. The number of tourists began to increase after 2016 when the offer was also adapted to the needs of individual tourists—now also including seniors and foreign tourists [40,42]. As many as 8000 tourists visited the mining village in 2018 and about 10,000 in 2021 (Figure 7) [42] and later oral interviews.

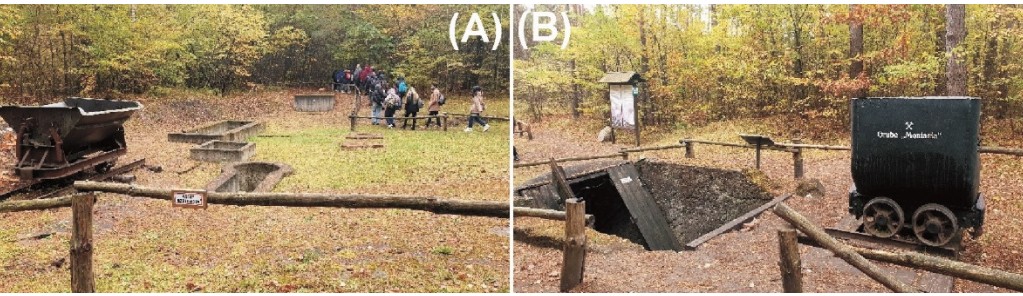

**Figure 5.** Mining relics after the „Montania" mine. (**A**)—coal washing tanks, (**B**)—dip heading.

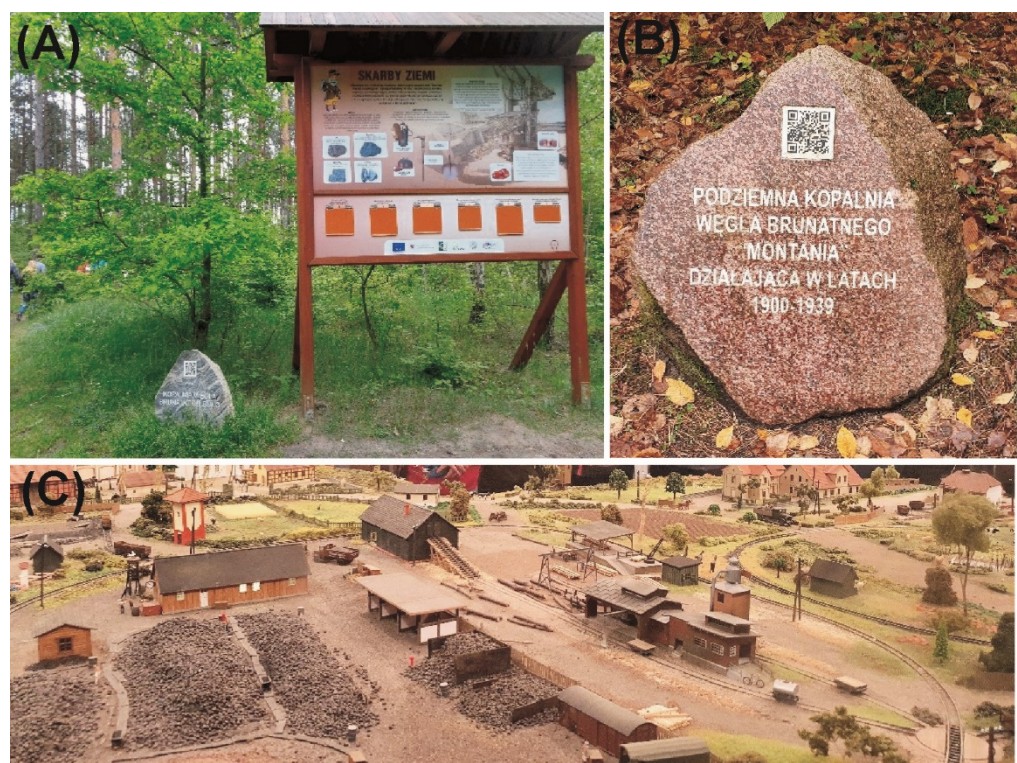

**Figure 6.** Educational elements in the mining village. (**A**)—educational board, (**B**)—"speaking stones", (**C**)—fragment of a model depicting the former mine.

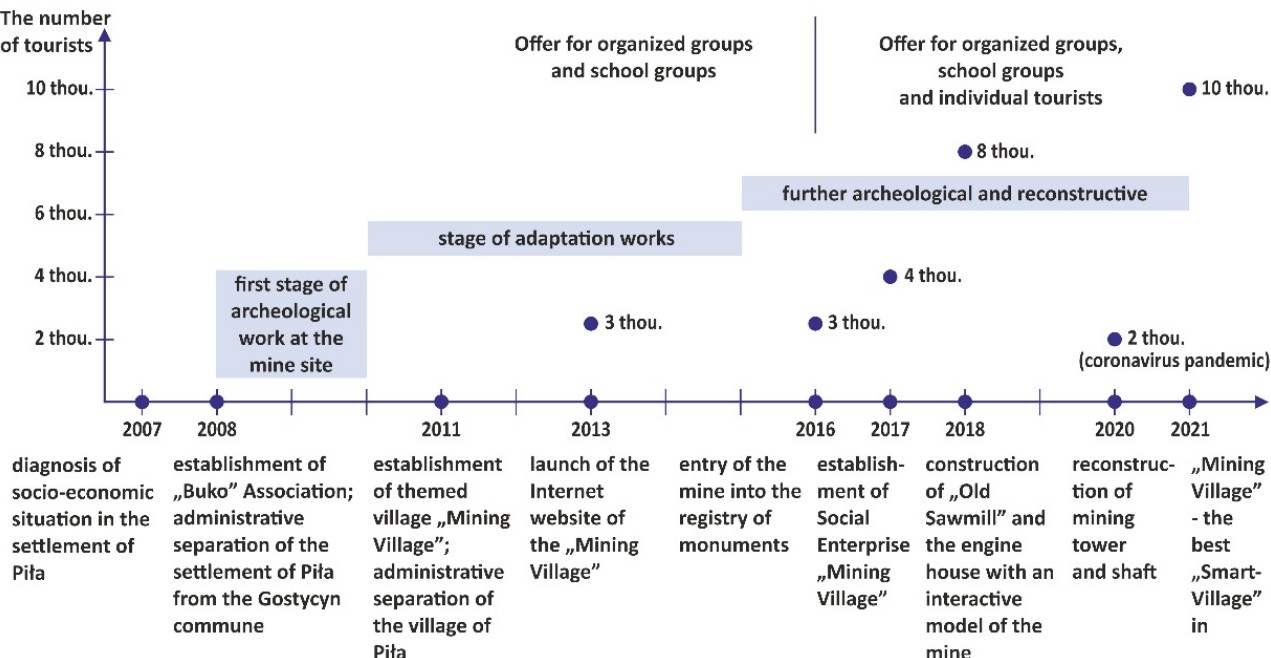

**Figure 7.** Most notable endeavours in the history of "Mining Village" and the number of tourists. Source: own elaboration based on [42] and oral interviews with W. Weyna.

## 4. Discussion

The conducted study indicates that cessation of brown coal extraction entails major changes in land relief. Most notably, the resulting land subsidence poses a threat to the local population, and thus, the sites of such occurrences need to be carefully analysed,

and post-mining areas constitute an activating factor to the local population. The cultural heritage is used for tourism-related purposes.

Laser scanning data allows capturing terrain relief changes in forestland. According to [44], airborne laser scanning allows analysing brownfields that are not marked on topographic maps, and in particular on historical maps. The above-mentioned authors discussed changes in the environment that followed the closing of underground brown coal mines in western Poland and established the course of underground galleries manifested as subsidence on the LIDAR image. The laser scanning area analysed in the present paper is presented very accurately in comparison to historical maps. Digital terrain models based on LIDAR data are very accurate, which allows identifying changes in the environment. Anthropogenic landforms are associated with the impact of the mines in operation and after decommissioning. Similar processes inducing environmental changes can also be observed in other regions of Poland, for instance in the Lubusz Region, where subsidence pits related to brown coal mines are still being formed [45].

Cessation of coal extraction and decommissioning of mines exerts considerable socio-economic influence. This is related to the loss of jobs, increase in unemployment and forced retraining of former miners [46]. The abandoned post-mining infrastructure (buildings, streets etc.) and excavations constitute an instance that can be used to improve the economic situation of the local population. This may positively influence the development of industrial and mining tourism [47,48]. Post-mining areas can be potentially used for establishing tourist trails in several countries [48]. The provided examples serve to show the potential of repurposing the mines in Piła Młyn for tourism. There is already a growing Interest in this form of tourism. This is related to the improvement in the economic situation of the local population.

In the case of industrial areas, tourist products need to be developed from the ground up. Actions need to concentrate on emphasizing selected attributes of the industrial environment and using them as a basis for establishing tourist attractions [49]. Archaeological and mining sites related to old mining operations can be divided into two kinds: above-ground sites that demonstrate characteristic landforms and sites found below the surface, i.e., underground mines. It is increasingly often argued that the relics of old mining operations should be preserved as they constitute objects of material culture of many peoples, nations and communities [50]. Moreover, such places are believed to hold considerable potential that can be tapped into to establish new industrial tourist products, which attract ever growing number of tourists [51–53]. These products not only constitute original tourist attractions, but also serve to activate places that have previously not been related to tourism. This is apparent in the case of themed villages whose development oscillates around a particular "theme", making them a unique tourist attraction [54]. Such villages strive for social and economic activation of the inhabitants, heritage preservation as well as development of education regarding cultural heritage [55].

The mining village has attracted tourists interested in industrial heritage and provided a basis for developing industrial tourism in the Tuchola Forest. Such tourism is developed making use of the cultural landscape of the former coal mines near Piła-Młyn. Interest in heritage normally focuses on coal mines in Lower Silesia [56] and Upper Silesia [57] and various kinds of post-industrial buildings that currently perform new, most often commercial, functions [58], technology trails [50] and museums [51].

One should note that the area of the Buko mine is not so much a degraded post-industrial district as it is historically forsaken. An example of another site in the Kuyavian-Pomeranian voivodeship that fell into oblivion is the salt mine in Inowrocław (Steinsalzbergwerk Inowrazlaw) [59]. Nowadays, this mine is not visible in the panorama of the town and the inhabitants do not associate it with an important part of the town's history and mining heritage. The last headframe of the mine was removed in the mid-1990s. The Bastion Association [59], willing to restore historical significance to the mine, postulates the need to disseminate knowledge about the mine, raise the interest of the city authorities and create an educational trail based on the relics of the mine. The mining village in Piła-Młyn

is an example of good practices, which is highlighted by, among other authors, [40] who saw the mining village as a source of inspiration to create a project of a tourism product called the Tarnów Lakeland that is also intended to integrate mining heritage with the natural environment and elements of the local culture.

There is currently little information on the layout of underground tunnels and very few witnesses that could provide their account of the situation and events preceeding the decommission of the mine. The available sources do not expressly indicate the location of all the mines (lack of historical maps).

## 5. Conclusions

Due to the absence of information about other underground lignite mining sites in northern Poland, the mining village in Piła-Młyn offers prospects for tourism development. It constitutes a good example of development of post-mining areas.

Historical topographic maps and LIDAR data illustrating the layout of respective galleries invisible on topographic maps are extremely helpful in recreating changes in terrain relief. This area features sediment pits that are an inevitable element of coal mining. It should be added that they can also pose a hazard, so careful exploration of the area is advisable.

All the time research, reconstruction and development works continue within the premises of the mine in order to recreate and conserve the monument in the form of old coal mines. It impacts the dissemination of knowledge about the cultural heritage of the Tuchola Forest and its previously unfamiliar history related to the mining industry. Such a past provided a basis for creating an interesting, genuine tourism product based on the integrity of the cultural, anthropogenic and natural environment. The fast-developing Mining Village cultivating mining traditions and offering additional attractions is a genuine tourism product in all Poland. The village is now an attraction that expands the tourist offer of the Tuchola poviat and belongs to the group of most important tourist destinations in the Kujawsko-Pomorskie voivodship. It stands as an example of good practices (winning many awards) in the context of grassroot initiatives, establishing and functioning of themed villages as well as preservation and promotion of post-mining relics. The innovative solutions implemented in 2021 there earned the mining village the status of a smart-village.

Post-mining areas should be carefully studied in terms of how they can be used to the benefit of the community. The occurrence of land subsidence may reveal a network of former mining corridors. This bears admittedly adverse consequences to residential development, but it also creates a considerable opportunity for the the development of tourism.

Interpretation of LIDAR data shows that the history of this area associated with brown coal mining has not been explored in full. Subsidence forms apart from those related to the presented coal mines are visible on the images. These will be the subject of future analyzed of the transformation of the terrain relief as a result of the end of lignite mining.

**Author Contributions:** Conceptualization, M.R. and A.G.; software, M.R. and A.G.; writing—original draft preparation, M.R., A.G. and M.H.; writing—review and editing, M.R., A.G. and M.H.; visualization, M.R. and A.G. All authors have read and agreed to the published version of the manuscript.

**Funding:** This research and APC were funded by the Project Supporting Maintenance of the Research Potential of the Institute of Geography at the Kazimierz Wielki University (grant number BS/2016/N1).

**Institutional Review Board Statement:** Not applicable.

**Informed Consent Statement:** Not applicable.

**Data Availability Statement:** The data presented in this study are available on the request from the corresponding author.

**Acknowledgments:** We would like to express our gratitude to Agnieszka Weyna and Wojciech Weyna from the "BUKO" Association for their valuable contribution during the writing of this work.

**Conflicts of Interest:** The authors declare no conflict of interest.

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
