# Peer review of "Environmental and Socio-Economic Effects of Underground Brown Coal Mining in Piła Młyn (Poland)"

_land, doi:10.3390/land11020219_

Round 1

Reviewer 1 Report

The article is correctly structured. The authors introduce readers to the issue of lignite mining in Poland, providing various methods of extraction and describing its impact on the environment. The work uses information from certain sources such as www.geoportal.gov.pl or www.baza.pgi.gov.pl.

Using LIDAR laser scanning and the available data, the authors describe with great accuracy the impact of extraction on the relief. The section "The use of coal mines for tourism" may be interesting for readers, as it presents the subject of research in terms of tourism.

In general, the article concerns a fairly narrow topic, which is tourism related to lignite mining. This article may be interesting for readers interested in this topic, especially the lack of connection between Bory Tucholskie and coal mining. In my opinion, the article has a rather overview character, which, however, fits in with the "Aims and Scope" of the "Land" journal.

I believe that the article can be published, my only objection is the quality of figure 5, it must be corrected before publication.

Author Response

Dear Reviewer

Thank you very much for your review. We corrected the remarks indicated in the text, as well as the figures. We have also responded to the reviewer's comments. The answers are below.

Answer:

The work contains a thread related to tourism, but the main theme is changes in the surface of the land. This is not the only example of changes in this area, as LIDAR has provided additional information on the relief that will be analyzed further.

Figure 5 has been corrected.

Best regards

Authors

Reviewer 2 Report

The work deals with an important and interesting problem concerning the environmental and socio-economic consequences of underground lignite mining. There is very little work on this subject, as this raw material is mostly mined open-pit. I think that the work is interesting, it brings new information to the subject, but the authors did not avoid shortcomings, which are listed below.

The drawings are made with care and they are very aesthetic, it is not clear to me where the studied area is located on the historical maps, I cannot find it in the description for the drawing, this figure requires improvement

Subsidence caused by mining in this part of Europe has been discussed in at least several studies, however, the authors do not refer to any of them.

The biggest drawback of the work is the poor literature review. There is a lot of work on the effects of underground exploitation of raw materials in this part of Europe and in other parts of the world. Some of the works that I mean are also based on the analysis of archival and contemporary cartographic materials and raise the issue of their suitability for the assessment of changes in the natural environment under the influence of the exploitation of raw materials, e.g.

Xiao, W., Hu, Z. & Fu, Y. Zoning of land reclamation in coal mining area and new progresses for the past 10 years. Int J Coal Sci Technol 1, 177–183 (2014). https://doi.org/10.1007/s40789-014-0024-3

Harnischmacher S, Zepp H (2014) Mining and its impact on the earth surface in the Ruhr District (Germany). Z Geomorph, Suppl Issues 58(3):3–22

Terrone, M.; Piana, P.; Paliaga, G.; D’Orazi, M.; Faccini, F. Coupling Historical Maps and LiDAR Data to Identify Man-Made Landforms in Urban Areas. ISPRS Int. J. Geo-Inf. 2021, 10, 349. https://doi.org/10.3390/ijgi10050349

SzypuÅ‚a, B. Digital adaptation of the Geomorphological Map of Upper Silesian Industrial Region, Poland (1:50,000)—Old map new possibilities. J. Maps 2020, 16, 614–624.

Henselowsky, F.; Rölkens, J.; Kelterbaum, D.; Bubenzer, O. Anthropogenic relief changes in a long-lasting lignite mining area (‘Ville’, Germany) derived from historic maps and digital elevation models. Earth Surf. Process. Landf. 2021, 46, 1725–1738.

Solarski, M., 2013. Anthropogenic transformations of the Bytom area relief in the period of 1883-1994. Environmental & Socio-economic Studies, vol. 1, issue 1. p. 1-8. DOI:10.1515/environ-2015-0001.  

 Dulias, R., 2016. The Impact of Mining on the Landscape, 351 A Study of the Upper Silesian  Coal Basin in Poland, Environmental Science and Engineering, Springer International Pub353

lishing Switzerland, pp 209.

SzypuÅ‚a, B.. "Spatial distribution and statistic analysis of the anthropogenic line forms on the different basic fields" Environmental & Socio-economic Studies, vol.1, no.2, 2015, pp.1-14. https://doi.org/10.1515/environ-2015-0007

Wójcik, J. (2013). Mining changes on the example of the WaÅ‚brzych basin relief (The Sudetes, Poland). Zeitschrift für Geomorphologie, 57(2), 187–205. https://doi.org/10.1127/0372-8854/2012/0090 

Kacper Jancewicz, Andrzej Traczyk & Piotr MigoÅ„ (2021) Landform modifications within an intramontane urban landscape due to industrial activity, WaÅ‚brzych, SW Poland, Journal of Maps, 17:4, 194-201, DOI: 10.1080/17445647.2020.1805805

Bauer, H.J. (1971) Recultivation and renewal of a balanced landscape in the lignite mining area of the Rhineland. Geoforum, 8(4), 31– 41. https://doi.org/10.1016/0016-7185(71)90028-5

Dickmann, F. (2011) Reclamation conditions of opencast mining in the Rhenish lignite-mining region (Germany). Zeitschrift für Geomorphologie, 55(1), 15– 24. https://doi.org/10.1127/0372-8854/2011/0055S1-0034

Feng, Y., Wang, J., Bai, Z. & Reading, L. (2019) Effects of surface coal mining and land reclamation on soils properties. A review. Earth-Science Reviews, 191, 12– 25. https://doi.org/10.1016/j.earscirev.2019.02.015

I think that 29 items cited are definitely not enough. I think it is worth referring to the works I have mentioned and to some other works related to this subject. The uniqueness of underground coal mines and their differences in environmental impact between this type of mining and underground mining should also be emphasized.

I believe that the work can be published after introducing the proposed corrections.

Author Response

Dear Reviewer

Thank you very much for your review. We corrected the remarks indicated in the text, as well as the figures. We have also responded to the reviewer's comments. The answers are below.

Answer:

We have added a paragraph with a detailed description of the indicated literature. Besides, we corrected the indicated figure. We supplemented the descriptions in the figures' titles. Now everything should be clearer.

Best regards

Authors

Reviewer 3 Report

I think that the title of this article should be slighly changed. The article mentions geotourism quite lavishly - perhaps it makes sense to mention it in the title?

Line 18. What is BUKO Association?

Lines 103-104: what is "[11-13 after 10,14-15]."?

In general, after editing, the article can be recommended for publication; I see no need for a second review.

Author Response

Dear Reviewer

Thank you very much for your review. We corrected the remarks indicated in the text, as well as the figures. We have also responded to the reviewer's comments. The answers are below.

I think that the title of this article should be slighly changed. The article mentions geotourism quite lavishly - perhaps it makes sense to mention it in the title?

Answer:

The work contains a thread related to tourism, but the main theme is changes in the terrain relief. This is not the only example of changes in this area, because LIDAR has provided additional information on the relief that will be analyzed further.

Line 18. What is BUKO Association?

Answer: BUKO is an abbreviation of the name of the association. We have completed this in the text.

Lines 103-104: what is "[11-13 after 10,14-15]."?

Answer: We changed it to the correct citation.

Best regards

Authors

Reviewer 4 Report

This study investigated the environmental and socio-economic effects of discontinuing lignite mining. Based on the results, changes in the environment are manifested in terrain relief and highlight the location of former coal mines. The current version of the manuscript is weak and needs further enrichment and improvement in terms of structure and theoretical aspects. Also, the Results, Discussion and Conclusion sections are not strong and fail to highlight the main implication of the findings. Overall, the current version has no capacity to be considered for publication in the journal of Land.

Author Response

Dear Reviewer

Thank you very much for your review. We corrected the remarks indicated in the text, as well as the figures. We have also responded to the reviewer's comments. The answers are below.

Answers:

Abstract:

  1. The English grammar and style should be checked throughout the manuscript.

Answer: The article was translated and revised by a native speaker.

  1. The authors should mention the main aim of the study in the Abstract section.

Answer: We changed that sentence.

  1. I miss more emphasis on the main significance of this study in Abstract. I suggest highlighting the main significance of the study in 1-2 sentences.

Answer: Sentence added.

  1. The authors should mention a few policy implications after the main recommendation based of results at the end of the Abstract in 1-2 sentences.

Answer: Sentence added.

  1. The authors should avoid repeating keywords already exists in the title (e.g. underground brown coal mine). The authors should replace them with new relevant words in the text.

Answer:  We have changed the indicated example.

Introduction

1.The Introduction section should be enriched by adding and citing several recent references (i.e. 2017- 2021). Also, the old references (1976-2005) should be replaced with the recent ones in the Introduction section as well as the whole manuscript.
Answer: We have extended the introduction to include the description of other works with similar topics.

  1. The Introduction section is too short (half of the page) and just briefly discussed the history of the opencast mining in Poland.

Answer: We have extended the introduction to include the description of other works with similar topics.

  1. Lines 50-51 in page 2: “These are Adamów, Koźmin, BeÅ‚chatów – Pole, BeÅ‚chatów, BeÅ‚chatów – Pole Szczerców, PÄ…tnów IV, Drzewce, TomisÅ‚awice, Turów and Sieniawa 2 [2].Ë® What do the authors mean by this sentence?

Answer: These are the Polish names of the lignite opencast mines operating in Poland at present.

  1. The authors should broadly discuss about socio-economic and environmental aspects of brown coal mining.

Answer: We added a paragraph in the introduction

  1. The introduction section needs to be enriched significantly. There are not promising numbers of recent and relevant references supporting the arguments. Moreover, there is not sufficient explanations on the effects of socio-economic factors on underground brown coal mining. The authors should discuss the concern of the present and future impact of underground brown coal mining on the natural environment.

Answer: This place is developing in terms of tourism, so one should expect an improvement, for example, in the tourist infrastructure. We will continue to observe this place.

  1. In the Introduction section, there should be a paragraph discussing the global novelty of the study comparing with previous studies. This is very important to first identify the gap in the previous studies, and then highlight how the current study is going to fill it

Answer: We added a sentence

Methodology

  1. Lines 60 in page 2: The authors should modify the section “2 Study area and MethodsË® to “2. MethodologyË®.

Answer: We changed it

  1. Line 61 in page 2: The authors should modify the subsection “2.1 Location and geologyË® to “2.1 Study areaË® and “2.2 Geological featuresË®.

Answer: We changed the title of this point but did not split it, beacouse there is not much information about the geological structure here.

  1. Lines 61-62 in page 2: “The study site is situated in Kuyavian-Pomeranian voivodeship in the village of PiÅ‚a MÅ‚yn (Fig. 1).Ë® When the authors want to introduce the study area should specify the location in the city, province and country and show it in the credible map. Therefore, the authors should revise this sentence based on the comment.

Answer: We changed it

  1. The authors should add geographical and demographic information of the study area.

Answer: It is a small village with not many inhabitants and a lot of information on other topics.

  1. Line 74 in page 3: “Figure 1. Location of the research area against the background of Poland and the Kuyavian-Pom-eranian Voivodeship (hypsometric map)Ë® is not a proper map to illustrate the study area location. Also this map is without legend and direction.

Answer: The legend is described in the title of the figure. The figure was corrected. Beside, in next point is a detailed map of the research area.

  1. Lines 78-80 in page 3: “The environmental consequences were presented using LIDAR (Light Detection and Ranging) data, (www.geoportal.gov.pl) [7], archival maps and the results of field studies.Ë® The authors should remove the link and cite the reference in the sentence. Also, address this comment in lines 82 and 84.

Answer: Corrected.

  1. Lines 98-131 in page 4: Inputs in these lines are related to the subsection of “3. History of brown coal mines in PolandË® and the authors should move them in the Introduction section.

Answer: This description concerns the history of lignite mines in Poland and is part of the research on the available literature.

  1. Figure 2 in page 5 is not a clear map. Also, this figure is without legend, grid, scale and direction. Therefore, the authors should modify it.

Answer: Corrected. The legend is in the title of the figure.

Results

  1. Inputs in lines 13-144 are related to environmental impact of discontinuing brown coal mining and more suitable for Introduction section.

Answer: This is an introductory paragraph to the description of changes in the environment.

  1. Line 255 in page 8: The authors should add numbering for subsection of “Amenities and tourist traffic in the Mining VillageË®

Answer:  We added number of Subdivision.

  1. Lines 250-253 in page 8: “Another step towards stimulating the inhabitants to action and improving the operation of the Mining Village was the establishment of a social enterprise called “Mining Village” that took over the service of tourist traffic and employed five professionally inactive inhabitants of the commune.Ë® The authors should reword, reformulate and split this sentence.

Answer: We changed it

18 Lines 253-256 in page 8: “The Mining Village plans to set up an inter-active underground tourism trail presenting the 19th century lignite mining technology in an innovative manner, a museum of brown coal mining in the Tuchola Forest and a centre of education on the conservation of monuments, culture and local history.Ë® The authors should avoid to write such too ling and heavy sentences.

Answer: In this sentence, we indicate what is planned to be done. We did not change this opinion.

  1. The resolution of the Figure 5 is too low and the authors should modify it.

Answer: We corrected this figure.

  1. Overall, the Results section like more a report of the study area and mining activities there. I didn’t find significant point in this section.

Answer: On Figure 5, which is based on oral information, shows the tourist movement. In addition, in the result part, we describe what has been done in this area.

Discussion

  1. The Discussion section is too short (just 2 paragraphs) and repeated what was discussed in the Results section.

Answer: We relate to works that take into account LIDAR and other effects of the mines' impact on the environment.

  1. In this section, the authors should summarize your results and outline their interpretation in light of the published literature. Then, please explain the importance of your results and finally acknowledge the shortcomings of the study.

Answer: We have added a paragraph with the description of the available literature in the introduction and in discusion.

Best regards

Authors

Reviewer 5 Report

The authors have presented a manuscript, describing the environmental and socio economic effects of underground brown coal mining in a region Poland. Following, I have included some comments to improve the manuscript.

  1. I suggest to the authors to add a new section detailing the state of the art. In this section, authors have to describe the relevant related work in which explain.
  2. Can the authors include at the end of the introduction, more details of the objectives of their study.
  3. This work presents very interesting results and practice to increase the turism in this region of Poland. I think that the authors can improve the format of results demonstration. The authors can highlight better the importance of the results obtained.
  1. Conclusions Consider extending the conclusions and adding a Future works paragraph. The summary and Conclusions, it is better to combine them in only section of conclusions.

Finally, the review is interesting and presents considerable information on the possibilities of the Brown coal mining in a region Poland and its possibilities for supplementation effects, but authors must improve the presentation of their results, discussion and conclusion. The topic in interesting, but the study lacks more details with precision, and concrete conclusion that will help farmers to improve or to change their strategies.

Author Response

Dear Reviewer

Thank you very much for your review. We corrected the remarks indicated in the text, as well as the figures. We have also responded to the reviewer's comments. The answers are below.

I suggest to the authors to add a new section detailing the state of the art. In this section, authors have to describe the relevant related work in which explain.

Answer:

Following another reviewer's pointers, we've added a paragraph in the introduction that describes more of a similar study.

  1. Can the authors include at the end of the introduction, more details of the objectives of their study.

Answer: We completed it.

  1. This work presents very interesting results and practice to increase the turism in this region of Poland. I think that the authors can improve the format of results demonstration. The authors can highlight better the importance of the results obtained.

Answer:  We are not currently expanding our results. We use the information available from the BUKO association.

  1. Conclusions Consider extending the conclusions and adding a Future works paragraph. The summary and Conclusions, it is better to combine them in only section of conclusions.

Answer: We have added future work on this topic in our conclusions, because as the area is of geological and geomorphological interest.

Finally, the review is interesting and presents considerable information on the possibilities of the Brown coal mining in a region Poland and its possibilities for supplementation effects, but authors must improve the presentation of their results, discussion and conclusion. The topic in interesting, but the study lacks more details with precision, and concrete conclusion that will help farmers to improve or to change their strategies.

Answer: At the moment, we have the data presented in the article. In the future, we will do research related to tourism. Beside, we added more information in Introduction.

Best regards

Authors

Round 2

Reviewer 4 Report

Thank you for the revised paper. The paper has been improved significantly in comparison to the previous version. However, I believe some issues need revision and clarification.

Author Response

Dear Reviewer

Thank you very much for your review. We corrected the remarks indicated in the text and figure. We have also responded to the reviewer's comments. The answers are below.

Answers:

Abstract

  1. Lines 17-20 in page 1: “To this end, historical topographic maps were used together with data from LIDAR (Light Detection and Ranging) laser scanning, available descriptions and scientific articles about coal mines, and information from the local inhabitants and representatives of the BUKO” Association (the Association of Inhabitants and Enthusiasts of PiÅ‚a nad BrdÄ… “BUKO”).Ë® is too long and heavy; the authors should revise and split this sentence.

    We splited this sentence.
  2. I miss more emphasis on the main significance of this study in Abstract. I suggest highlighting the main significance of the study in 1-2 sentences.

    We added sentence.
  3. The authors should mention a few policy implications after the main recommendation based of results at the end of the Abstract in 1-2 sentences.

    We added sentence.

Introduction

  1. Lines 57-58 in page 2: “Extraction of hard and brown coal contributes to changes in the natural environment.Ë® The authors should explain in detail how extraction of hard and brown coal contributes to changes in the natural environment?

    We have these points detailed in Introduction in lines 58-72.

  2. Lines 66-68 in page 2: “The authors [7] reviewed available literature to demonstrate changes in physical, chemical and biological properties of soils in the vicinity of open-cast brown coal excavation sites.Ë® The authors should modify citation (authors [7] should be changed to Feng et al. [7]). Moreover, the authors should highlight main findings of this study.

    We changed it and we added main conclusion.

  3. Lines 82-83 in page 2: “Other authors [18] extended their analyses to account for linear element such as trenches and railroad embankments.Ë® The authors should add citations instead of mention “Other authors [18]Ë®.

    We changed it.

  4. Lines 84-85 in page 3: “Relief transformations tend to occur not only in the vicinity of decommissioned mining site, but also in rural and urban areas.Ë® is vague; the authors should rewrite it.

    We changed this sentence.

  5. Lines 97-98 in page 3: “The authors indicate collapsed beaver dens as one of such naturally occurring land relief elements.Ë® Again, the authors should explicitly add references not to mention “authorsË®.

    We changed citation.

  6. To the end of the Introduction section there should be a paragraph discussing the global novelty of the study comparing with previous studies. Also, there should be a few research questions linked to the main objectives.

    We added more information.

Methodology

  1. As I have mentioned in previous revision, the authors should add geographic and demographic information of the study area.

    We added this information.

  2. Description of the applied method is not enough and the authors should enrich it.

    We added few sentence.

  3. The section “3. History of brown coal mines in PolandË® is wrongly placed between Methodology section and Results section. But this section is more fit with the Introduction section. Hence, the authors should move it in the proper section.

         We added this section to section Results as a information about history brown coal mining in Poland.

Results

  1. Line 235 in page 9: “The first is 95 m wide.Ë® is too short sentence; please enrich it or merge with the next sentence.

         We merged this with next sentence.

  1. Lines 312-315 in page 13: “arts, cooking and nature, gardening, old-fashioned professions and workshops during which gala mining.Ë® This is an incomplete sentence and the authors should modify it. It is not our mistake.

    It is a part of bigger sentence and it will be correct in article in Word version.

  2. The resolution of the figure 7 is still too low and needs to be revised.

    We corrected this figure.

Discussion

  1. I miss a real discussion on the main results. The authors should 1) discuss the main findings, 2) compare/contrast them with the similar studies’ findings (i.e., start with discussing the main results of your study, then compare/contrast your findings with other relevant studies (2015-2022)), and 3) add limitations of your study in 1 paragraph.

    We added main results, similar studies and limitations. We hope than Discussion is better now.

Conclusion

  1. The authors should highlight the specific and practical suggestions with respect to their findings at the end of the Conclusion section in one paragraph.

    We added conclusion.

Best regards

Authors